# A Transcriptional Activator of Ascorbic Acid Transport in *Streptococcus pneumoniae* Is Required for Optimal Growth in Endophthalmitis in a Strain-Dependent Manner

**DOI:** 10.3390/microorganisms7090290

**Published:** 2019-08-24

**Authors:** Angela H. Benton, Mary Darby Jackson, Sandy M. Wong, Justine L. Dees, Brian J. Akerley, Mary E. Marquart

**Affiliations:** 1Department of Microbiology and Immunology, University of Mississippi Medical Center, Jackson, MS 39216, USA; 2Current affiliation: Virginia Tech, Blacksburg, VA 24061, USA

**Keywords:** ascorbic acid, endophthalmitis, *Streptococcus pneumoniae*

## Abstract

*Streptococcus pneumoniae* is among the top causes of bacterial endophthalmitis, an infectious disease of the intraocular fluids. The mechanisms by which *S. pneumoniae* grows and thrives in the intraocular cavity are not well understood. We used a bacterial genome-wide assessment tool (transposon insertion site sequencing) to determine genes essential for *S. pneumoniae* growth in vitreous humor. The results indicated that an ascorbic acid (AA) transport system subunit was important for growth. We created an isogenic gene deletion mutant of the AA transcriptional activator, *ulaR2*, in 2 strains of *S. pneumoniae*. Growth curve analysis indicated that *ulaR2* deletion caused attenuated growth in vitro for both strains. However, in vivo vitreous humor infection in rabbits with either strain determined that *ulaR2* was necessary for growth in one strain but not the other. These results demonstrate that *ulaR2* may be important for fitness during *S. pneumoniae* endophthalmitis depending on the background of the strain.

## 1. Introduction

Streptococcal species are among the top causes of bacterial endophthalmitis, an infectious disease of the aqueous and/or vitreous humors of the eye [1,2,3]. Exogenous bacterial endophthalmitis can result from contamination during penetrating events such as ocular trauma, cataract surgery, or intravitreal injections of medications for other ocular disorders [4]. Infections after intravitreal injections in particular appear to be strongly associated with streptococci, likely due to contamination from the oral cavity during the injections [5]. The prognosis of *S. pneumoniae* endophthalmitis is notably poor, with frequent outcomes of low visual acuity or loss of the eye [1,6,7].

The current knowledge of *S. pneumoniae* endophthalmitis includes demonstration that at least two bacterial virulence factors are important for disease. The cholesterol-dependent cytolysin of *S. pneumoniae*, pneumolysin, was shown to be necessary for ocular disease severity in animal models of endophthalmitis [8,9,10]. However, blocking pneumolysin by immunization had no effect on bacterial recovery from the vitreous humor [10]. The outer polysaccharide capsule of *S. pneumoniae*, which forms the basis of serotyping encapsulated strains and is classically associated with immune evasion, was also determined to be essential for virulence during endophthalmitis [11]. Ocular disease severity, as well as bacterial recovery from the vitreous humor, were significantly lower in rabbits infected with an isogenic capsule-deficient strain of *S. pneumoniae* compared to the parent strain.

One factor that has not been fully addressed is the growth of *S. pneumoniae* in the vitreous humor. Streptococci are fastidious organisms that ferment sugars and produce lactic acid, and are typically grown in low oxygen environments in rich media. Moreover, growth to mid-logarithmic phase from 100-fold dilution of stationary (overnight) cultures in vitro takes approximately four to 12 h depending upon the specific strain (unpublished observation). Vitreous humor of the rabbit eye appears to serve as a suitable growth medium for *S. pneumoniae*, with bacterial loads of 10^6^ colony-forming units (CFU) per mL to 10^10^ CFU/mL being recovered after inoculations of only 10^2^ CFU [10,11,12]. Glucose, the preferred carbon source for *S. pneumoniae*, is available in the vitreous humor at millimolar concentrations [13]. However, we previously showed that increased quantities of vitreous glucose did not affect bacterial load quantities [14]. Alternate carbon sources for *S. pneumoniae* in the vitreous include ascorbic acid (AA), hyaluronan, and sialic acid [15,16,17,18].

Examination of individual vitreous components is hindered by the quantity and complexity of those components [19,20,21]. Therefore, in order to investigate which factors impact the intraocular growth of *S. pneumoniae*, we employed transposon insertion site sequencing of a library of transposon mutants harvested after incubation in vitreous humor and compared the sequencing reads to those of the same library grown in bacteriological medium (Todd Hewitt broth with yeast extract). This technique identified *S. pneumoniae* genes that were essential for growth in the vitreous humor. We selected one of these genes, an ascorbic acid transporter subunit, to address the hypothesis that ascorbic acid transport is essential for growth of *S. pneumoniae* in vitreous humor. Targeted genetic deletion of a transcriptional activator of ascorbic acid transport in two strains of *S. pneumoniae* revealed that ascorbic acid transport may be essential in a strain-specific manner in the environment of the vitreous humor.

## 2. Materials and Methods

### 2.1. Bacterial Strains and Culture Conditions

*S. pneumoniae* D39, a well-characterized and commonly used laboratory strain of capsule type 2, was provided by Larry McDaniel at the University of Mississippi Medical Center, Jackson, MS, USA. *S. pneumoniae* E335, a capsule type 19F human endophthalmitis strain, was provided by Regis P. Kowalski at the Charles T. Campbell Eye Microbiology Laboratory, University of Pittsburgh, Pittsburgh, PA, USA. Frozen aliquots of a transposon library containing over 20,000 *S. pneumoniae* D39 mutants were generously provided by Andrew Camilli, Tufts University School of Medicine, Boston, MA, USA. Construction of this library was previously described [22]. *S. pneumoniae* TIGR4 containing a deletion of *ccpA* was provided by Andrew Camilli and served as the source of DNA template for a chloramphenicol resistance cassette [23].

D39 and E335 were maintained in Todd Hewitt broth containing 0.5% yeast extract (THY) plus 20% glycerol as frozen stocks. Frozen stocks were routinely cultured for isolation on sheep’s blood agar for 18–24 h at 37 °C and 5% CO_2_. Isolated colonies were inoculated into THY and incubated for 18 h at 37 °C and 5% CO_2_ prior to subculturing for experiments. D39 transposon library was grown from frozen stock in THY containing 200 µg/mL spectinomycin until the optical density at 600 nm (OD_600_) was 0.1. This culture was then used to seed fresh THY or naïve rabbit vitreous humor (Pel-Freez, Rogers, AR, USA) at a 100-fold dilution, which translated to an inoculation of approximately 10^4^–10^5^ CFU of the starting library. The inoculated THY and vitreous humor were incubated for 6 h at 37 °C and 5% CO_2_. Bacterial genomic DNA was harvested and purified from each environment (THY or vitreous humor) for subsequent preparation for sequencing. Despite the production of a bottleneck by this design (i.e., not all genes may be represented in the mutant bank), mutants containing disruptions of genes required for growth in vitreous humor will not survive, or will be significantly less abundant, in vitreous humor compared to their abundance in THY.

### 2.2. Preparation and Sequence Analysis of Transposon Libraries

D39 transposon library DNA following incubation in THY or rabbit vitreous humor was prepared according to published methods [22] and consisted of 10^4^–10^5^ individual transposon mutations in a collection pool of approximately 10^8^ CFU/mL for each environment (THY or vitreous humor). Briefly, bacterial DNA was purified from each environment after 6 h of incubation then digested with the restriction enzyme *Mme*I and ligated to adapter oligonucleotides (Illumina, San Diego, CA, USA) using T4 DNA ligase according to Basic Protocol 3 of van Opijnen et al. [22]. A PCR of 22 cycles with a 55 °C annealing temperature was then performed to amplify the 120-bp targets. The target PCR products were gel-purified and subjected to next-generation sequencing at the Tufts University Core Genomics Facility, Boston, MA, USA.

Sequencing reads containing the *magellan6* derivative of the *Himar1* transposon [22] arm sequence were trimmed and aligned to the D39 genome (GenBank accession #NC008533) with Bowtie 1.1.2 [24] using the “best only” setting with no mismatches. Reads and insertion sites within each gene were tabulated on both strands. Log-transformed average reads per gene for two independent biological replicates of each condition were normalized and compared by the Student’s *t*-test (two-tailed, unequal variance). THY-grown samples averaging 1.2 × 10^6^ total sum reads and vitreous-grown samples averaging 2.1 × 10^7^ total sum reads were obtained, collectively covering 298 genes.

### 2.3. Construction of ulaR2 Deletion Mutants

Genomic DNA from *S. pneumoniae* TIGR4 with *ccpA* deletion, D39, and E335 was purified by standard phenol: chloroform extraction methods. Splice overlap extension (SOE) PCR was used to construct deletion mutants of *ulaR2* in D39 and E335 [25]. The chloramphenicol resistance cassette (750 bp) was amplified from the TIGR *ccpA* deletion strain using the primers catF1 and NcatR1 as described [23]. Oligonucleotide primers were designed to amplify 800 bp flanking sequences on either end of *ulaR2* from genomic DNA of strains D39 (GenBank accession #NC008533) and E335 (GenBank accession #CP026670). The reverse primer of the upstream portion (spd1961_upR) contained additional sequences that overlapped with the beginning of the chloramphenicol resistance cassette, and the forward primer of the downstream portion (spd1961_downF) contained additional sequences that overlapped with the end of the chloramphenicol resistance cassette (Table 1). Phusion high-fidelity DNA polymerase (Thermo-Scientific, Waltham, MA, USA) was used for amplification according the parameters recommended by the manufacturer and with an annealing temperature of 51 °C.

The two flanking sequence PCR products (800 bp each) and the amplified chloramphenicol resistance cassette (750 bp) were gel-purified. Equal concentrations of each piece were mixed together and subjected to a final PCR amplification reaction using the forward primer of the upstream segment and the reverse primer of the downstream segment, resulting in a single PCR product of 2350 bp. This product was then gel-purified and used as donor DNA to transform *S. pneumoniae* D39 or E335 according to established methods [26]. Transformants were selected on blood agar containing 4 µg/mL chloramphenicol and were screened by PCR. Successful gene deletion mutants were confirmed by sequencing (MCLAB, San Francisco, CA, USA). Deletion mutants were designated D39Δ*ulaR2* and E335Δ*ulaR2*.

### 2.4. S. pneumoniae Pangenome Construction

The pangenome and core genome were determined for *S. pneumoniae* using the pangenome analysis pipeline Roary (https://sanger-pathogens.github.io/Roary/). Sixty-three complete and draft *S. pneumoniae* genomes that had protein annotation were downloaded from National Center for Biotechnology Information (NCBI) on January 25, 2018. The gbff files were converted to GFF3 files using the Bio:Perl script bp_genbank2gff.pl. The 63 converted files were then used for a pangenome analysis through the Roary pipeline. Roary was run a total of 5 times with a 90% blastp percent identity cutoff, resulting in 5 pangenomes being generated. The mode of the 5 pangenomes generated was used to represent the number of genomes in which each gene appears.

### 2.5. Growth Experiments in Vitro

D39, D39Δ*ulaR2*, E335, and E335Δ*ulaR2* were cultured on blood agar or blood agar containing chloramphenicol for approximately 24 h at 37 °C and 5% CO_2_. Isolated colonies were inoculated into pre-warmed THY or THY containing 4 µg/mL chloramphenicol. Cultures were allowed to grow statically for 14–18 h at 37 °C and 5% CO_2_, then were used to create subcultures at a 1/100 dilution in THY or naïve rabbit vitreous humor. Cultures were sampled at 4, 6, 8, 12, and 24 h for quantitation of growth by plating of serial dilutions. Three biological replicates of each strain and each condition were done. Each parent/mutant pair was compared by *t*-test with an assumption for unequal variance.

### 2.6. Growth and Virulence in Vivo

Sixty-four 2.4–2.8 kg specific-pathogen-free female New Zealand white rabbits were purchased from Charles River Laboratories (Oakwood Research Facility, Mattawan, MI, USA). Animals were maintained according to the tenets of the Association for Research in Vision and Ophthalmology Statement for the Use of Animals in Ophthalmic and Vision Research. This study was approved by the University of Mississippi Medical Center Institutional Animal Care and Use Committee (protocol #1093D). Rabbits were anesthetized with ketamine/xylazine and their eyes received topical drops of proparacaine prior to infection. Left eyes were injected intravitreally with 10^2^ CFU of logarithmic phase D39, D39Δ*ulaR2*, E335, or E335Δ*ulaR2*. The inocula were plated to verify the accuracy of the quantity of bacterial CFU. Right eyes were not manipulated. Eyes were examined by slit lamp biomicroscopy and assigned disease severity scores [27] 24 and 48 h after infection. Rabbits were euthanized with overdoses of sodium pentobarbital followed by bilateral pneumothorax while under anesthesia after the last examination. Vitreous humor was extracted, serially diluted, and plated on blood agar for the enumeration of bacterial loads. A subset of vitreous humor was tested for myeloperoxidase (MPO) activity to estimate polymorphonuclear neutrophil (PMN) activity according to the method of Williams et al. [28]. Flow cytometry of vitreous humor was done using a mouse anti-rabbit T cell and neutrophil isotype IgG1 monoclonal antibody (clone RPN3/57; Bio-Rad, Hercules, CA, USA) coupled to cytometric beads (BD Biosciences, San Jose, CA, USA) to detect PMNs. Animal experiments were split into two separate events for each parent and mutant strain pair to verify reproducibility of results. Slit lamp examination scores, myeloperoxidase activity quantities, and bacterial load recoveries from vitreous samples were analyzed using the Mann–Whitney *U* test.

## 3. Results

### 3.1. Transposon Mutant Library Analysis

Comparison of sequencing reads between the D39 transposon mutant library grown in THY and that grown in rabbit vitreous humor indicated 6 specific *S. pneumoniae* genes required for growth in vitreous humor (Table 2). *Spd_0095* and *spd_1225* encode conserved hypothetical proteins with unknown function. *Spd_0095* is 594 nucleotides in length and appears in 100% of the complete *S. pneumoniae* genomes available in GenBank. A National Center for Biotechnology Information (NCBI) BLASTx search indicated that the product of *spd_0095* is a membrane protein. *Spd_1225* is 1272 nucleotides and appears in 95% of complete *S. pneumoniae* genomes. Kyoto Encyclopedia of Genes and Genomes (KEGG; https://www.genome.jp/dbget-bin/www_bget?spd:SPD_1225) and BLASTx searches suggest this gene product belongs to the pyridoxal phosphate-dependent aminotransferase family proteins. The gene encoding transcription repair coupling factor (*spd_0006*) is involved in nucleotide excision repair. NADP-dependent glyceraldehyde-3-phosphate dehydrogenase, or GapN, is encoded by *spd_1004*. GapN is a glycolytic enzyme that catalyzes the oxidation of glyceraldehyde-3-phosphate to 3-phosphoglycerate in an irreversible manner and generates NADPH for energy. *S. pneumoniae* also possesses a classical GAPDH that functions similarly but generates ATP for energy. N-acetylneuraminate lyase, encoded by *spd_1489*, is an enzyme that catalyzes the catabolism of N-acetylneuraminate (sialic acid) to N-acetyl-D-mannosamine and pyruvate. The phosphotransferase system (PTS) subunit IIB, ascorbate transporter gene (*spd_1846*) is part of a 9-gene operon specific for the transport of ascorbic acid (AA) into the bacterial cell [29]. The expression of this operon is upregulated in the presence of AA [29]. The vitreous humor is known to contain AA [16,17]. Therefore, we hypothesized that AA transport would be essential for growth of *S. pneumoniae* in the vitreous and aimed to test our hypothesis using targeted deletion of a known AA transport gene.

### 3.2. Creation of ulaR2 Deletion in S. pneumoniae

Transport of AA in *S. pneumoniae* D39 is accomplished by 2 known transporters to date. *Spd_1846*, the gene identified in our transposon mutant screen as being required for growth in vitreous humor, is part of one operon (*ula*) for AA transport (Appendix A). Afzal et al. [29] determined that the *ula* operon was upregulated by ascorbic acid and identified *ulaR* as the regulator required for expression of this operon. However, the authors noted that there was no growth defect of a mutant deficient in *ulaR* compared to the parent strain when cultured in AM17 medium, a rich medium supplemented with ascorbic acid. We constructed a mutant of D39 with a deletion of *spd_1846*, replaced with a chloramphenicol resistance gene possessing a transcriptional stop codon, and observed no significant differences compared to parental D39 in growth in vitro, whether in THY or vitreous humor (data not shown). These results indicate more than one possible scenario. The transposon disruption of *spd_1846* could have caused a nonpolar effect on an unknown gene that reduced fitness. Alternatively, the transposon library mutants disrupted in *spd_1846* could have had reduced growth when in the presence of competing bacteria (other transposon mutants). However, another operon for AA transport, *ula2*, is also present and is controlled by *ulaR2*. It has been surmised that lack of growth defect in a *ulaR* mutant could be due to compensation by *ulaR2* [30]. We, therefore, selected *ulaR2* for targeted gene deletion in D39 and a clinical endophthalmitis strain, E335.

### 3.3. Pangenome Analysis

We performed a pangenome analysis of *S. pneumoniae* to identify the presence of the *ula2* operon, a 5-gene operon, in 63 strains with completed or draft genomes. *Spd_1957*, *spd_1959*, and *spd_1960* were found in 100% of the strains. *Spd_1958* and *spd_1961* were found in 70% and 60%, respectively. All 5 genes were verified to be present in D39 and E335. A summary of the pangenome analysis can be found in Appendix A. Appendix A shows the results of 1 of the 5 runs from Roary, and lists the presence or absence of each gene in each of the 63 genomes. Appendix A lists the assembly codes for each strain, which are the codes present in Appendix A. The *ulaR2* gene differs by 5 nucleotides between strains D39 and E335; these nucleotide changes do not result in amino acid sequence differences (Appendix A).

### 3.4. Growth of D39, D39ΔulaR2, E335, and E335ΔulaR2 in Vitro

Each strain was grown in THY or rabbit vitreous humor and sampled at selected times to quantitate CFU/mL. There were no significant differences between each parent and mutant pair when grown in THY at any time point except for E335 and E335Δ*ulaR2* at 12 h (Figure 1a). In contrast, CFU/mL were significantly higher (10- to 100-fold, or 1-2 log-fold) for D39 compared to D39Δ*ulaR2* at 6 through 24 h when grown in vitreous humor (Figure 1b). Likewise, CFU/mL were significantly greater (up to almost 100-fold or almost 2 logs higher) for E335 compared to E335Δ*ulaR2* at 8 through 24 h in vitreous (Figure 1c). These data suggest that deletion of *ulaR2* reduces fitness of *S. pneumoniae* in vitreous humor.

### 3.5. Growth and Virulence of D39, D39ΔulaR2, E335, and E335ΔulaR2 in Vivo

The left eyes of each rabbit were intravitreally injected with 10^2^ CFU of bacteria. Each of the 4 strains had *n* = 8 eyes for a 24-h endpoint and *n* = 8 eyes for a 48-h endpoint. Enumeration of the inocula by serial dilution and colony counts determined the actual inocula to be 137 and 142 CFU for D39, 167 and 145 CFU for D39Δ*ulaR2*, 77 and 160 CFU for E335, and 57 and 117 CFU for E335Δ*ulaR2*. Animals infected with D39 and D39Δ*ulaR2* had no significant differences in mean slit lamp examination scores [27] at 24 or 48 h after infection (Figure 2). E335 and E335Δ*ulaR2* also were not significantly different in scores. The mean bacterial loads recovered from vitreous humor of rabbits infected with D39 were significantly higher (over 10-fold) than those infected with D39Δ*ulaR2* at both 24 and 48 h after infection (Figure 3). In contrast, bacterial loads were not significantly different between E335-infected and E335Δ*ulaR2*-infected eyes at either time point; however, E335Δ*ulaR2* did not show increased fitness compared to E335.

Vitreous humor MPO activity was significantly higher (approximately 100-fold or 2 log-fold) in D39-infected eyes compared to eyes infected with D39Δ*ulaR2* at 48 h, suggestive of higher PMN activity in those eyes (Figure 4a). However, the quantity of PMNs as determined by flow cytometry trended to be higher for eyes infected with the mutant strain 48 h after infection, although not significant (Figure 4b). Taken together, these data indicate that more PMNs may be present in eyes infected with the mutant, but that those PMNs are less active. If the PMNs are more active in D39-infected eyes yet the bacterial loads are also higher, either the wild type bacteria are causing more activation of PMNs or the wild type bacteria could be escaping opsonophagocytosis by an undetermined mechanism that is not occurring for the *ulaR2* mutants. Vitreous humor MPO activity was also assessed for E335 and E335Δ*ulaR2*; however, there were no significant differences in MPO activity between eyes infected with E335 and E335Δ*ulaR2* (Figure 4a). Sample quantities were insufficient to also perform PMN quantitation by flow cytometry for vitreous infected with E335 and E335Δ*ulaR2*.

## 4. Discussion

The results of this study illustrate the growth and virulence differences between strains of *S. pneumoniae*. We hypothesized that AA transport would be necessary for growth in vitreous humor following the utilization of a D39 transposon mutant library to conduct negative selection screening. Subsequent results using a targeted gene deletion of a regulator of AA transport, *ulaR2*, supported our findings from the transposon library in D39, but not when we tested the same targeted deletion in an endophthalmitis isolate, E335. Although *ulaR2* differs by 5 nucleotides between D39 and E335 (Appendix A), the translation is unchanged. A limitation of this study that may have an effect on the differences noted for D39 and E335 is the use of a single time point for initial screening of the negative selection with the transposon library. Additional time points could reveal information regarding fitness in the vitreous humor that was not captured previously. We were also unable to create complemented strains of the mutants; many *S. pneumoniae* studies testing isogenic mutations appear to lack complemented strains [29,30,31,32,33]. Nevertheless, our mutants were generated in frame and were verified by sequencing to be located in the correct position.

The in vitro growth experiments (Figure 1) indicated that deletion of *ulaR2* in E335 caused reduced growth in vitreous humor, yet there was no significant reduction in the subsequent animal experiments (Figure 3). This finding suggests that true physiological conditions play an important role in bacterial growth and virulence. We hypothesize that replenishment of vitreous components that occur during daily turnover is involved in this pathogen-host interaction. The vitreous is also an excellent buffer, as we have observed only small decreases in pH after 24 h of bacterial growth in vitro (unpublished observations). Since the buffering capacity of the rabbit vitreous appears to remain intact ex vivo, it is likely that there are additional factors involved in the difference between bacterial growth in vitro and in vivo. The buffering capacity could play a role in the in vitro growth differences between THY and vitreous humor (Figure 1), especially with regard to E335. Regardless of the *ulaR2* mutation, E335 did not survive in THY beyond 24 h of growth, yet was still viable in vitreous humor. We observed that the pH of THY dropped from neutral to approximately 5.5 by 24 h of growth in vitro, which could explain the growth difference. Alternatively, THY is a complex and rich medium that is not minimally defined; identification of key differences between THY and vitreous humor may indicate specific nutrients provided by vitreous that are absent in THY.

AA is one of the more than 30 different carbohydrates *S. pneumoniae* has been shown to metabolize in vitro [18,29,34]. Similar to our findings, AA transport was determined to be important for *S. pneumoniae* in the nasopharynx [35]. Three genes in the *S. pneumoniae* TIGR4 genome (*spn_2031*, *spn_2033*, and *spn_2036*) were found to be essential for bacterial infection and survival. All 3 of these genes were “responsive in vivo,” meaning the lack of the gene resulted in a significant change in phenotype in the nasopharynx [35]. *Spn_2036* of TIGR4 has 99% identity to *spd_1845* (*ulaA* of the *ula* operon) of D39. *Spn_2031* has 99% identity to *spd_1840* (*ulaG* of *ula*), which transcribes an L-ascorbate 6-phosphate lactonase. *Spn_2033* has 99% identity to *spd_1842* (*ulaF* of *ula*), which transcribes an L-ribulose-5-phosphate 4-epimerase. All 3 TIGR genes also had 99% identity to the corresponding genes in E335.

The genome of D39 is 2,046,115 bp and has 1986 genes whereas E335 is 2,221,316 bp and has 2254 genes (Appendix A). Although these two strains possess the five genes of the *ula2* operon as determined by sequence analysis, their growth phenotypes differed upon *ulaR2* deletion. Comparison of the presence and absence of specific genes in D39 and E335 (Appendix A) shows that there are additional putative carbohydrate transport genes, as well as phage genes, present in E335 that are not present in D39. Bioinformatic analysis of the key differences between the genomes of these strains could yield important information as to the robustness of E335 in the vitreous humor and aid in identifying *S. pneumoniae* environmental adaptations. With regard to the host response, questions remain as to how the bacteria escape killing by neutrophils and what factors are responsible for eliciting the influx of the neutrophils.

A question that remains is whether the fitness of *S. pneumoniae* (or other endophthalmitis-causing bacteria) is similar in rabbits and humans. The human vitreous proteome contains over 1000 unique proteins [20], therefore, pinpointing the specific similarities and differences between mammalian species could be challenging. One general similarity is that *S. pneumoniae* endophthalmitis in humans is associated with a poor prognosis [1,6,7], and infection of rabbit vitreous humor with human endophthalmitis strains results in severe endophthalmitis [10,12]. A previous report of a *Staphylococcus aureus* transposon mutant library in bovine intraocular fluids identified genes essential for fitness that are integral to glycolysis and gluconeogenesis [36]. Their findings combined with our identification of GapN and an AA transport subunit (Table 2) highlight the vitreous humor as a rich source of carbohydrates that can support bacterial growth. One of the keys to potentially inhibit that growth will be to dissect which carbon sources can be targeted for certain species, and to determine whether the endophthalmitis-adapted strains can be inhibited as a whole.

## Figures and Tables

**Figure 1 microorganisms-07-00290-f001:**
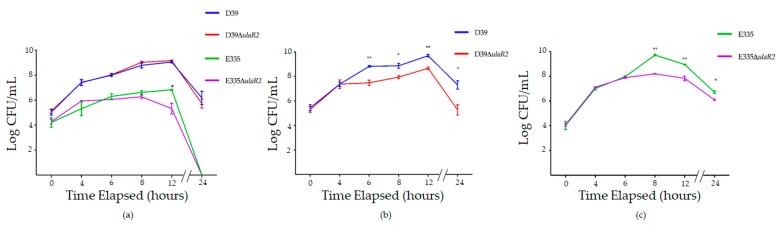
Growth of D39, E335, and corresponding *ulaR2* isogenic mutants in vitro. (**a**) Growth of all strains in THY; (**b**) growth of D39 and D39Δ*ulaR2* in rabbit vitreous humor. (**c**) Growth of E335 and E335Δ*ulaR2* in rabbit vitreous humor. Bacterial growth is expressed in Log_10_ units. Asterisks indicate the following: * *p* < 0.05, ** *p* < 0.01. Error bars represent standard errors of the means.

**Figure 2 microorganisms-07-00290-f002:**
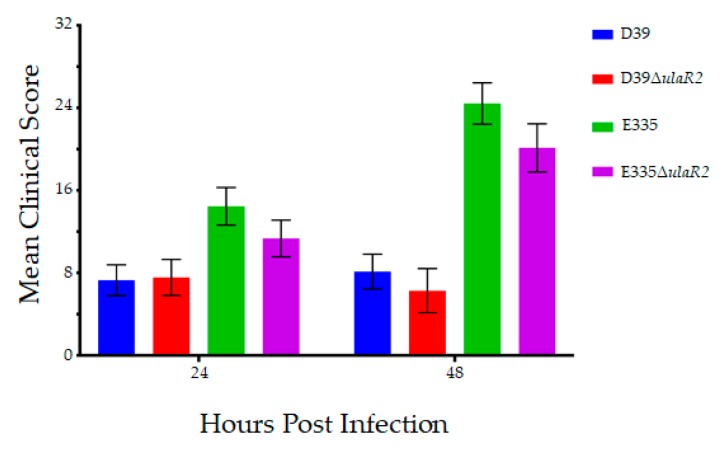
Mean clinical scores of eyes 24 and 48 h after infection. Eyes infected with each parent strain and its corresponding isogenic *ulaR2* mutant strain produced clinical scores that were not significantly different from each other. Error bars represent standard errors of the means.

**Figure 3 microorganisms-07-00290-f003:**
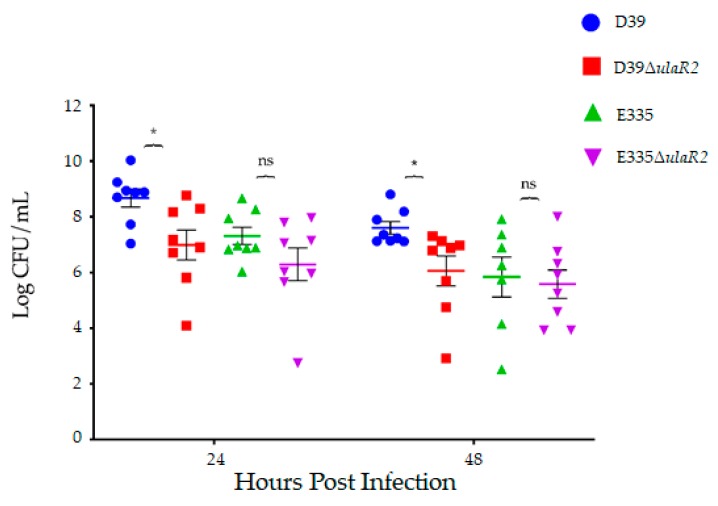
Bacterial recovery from rabbit vitreous humor infected with each strain 24 and 48 h after infection. Asterisks indicate the following: * *p* < 0.05. “ns” indicates “not significant.”

**Figure 4 microorganisms-07-00290-f004:**
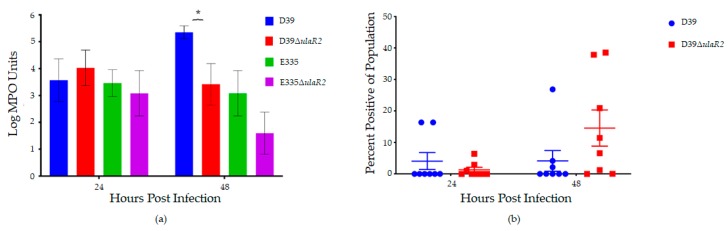
Polymorphonuclear neutrophil (PMN) activity and presence in infected vitreous humor of rabbits. (**a**) Mean Log_10_ MPO units of activity calculated from spectrophotometric absorbance measurements. Asterisk indicates *p* < 0.05. Error bars represent standard errors of the means; (**b**) percent of the total population of cells in infected vitreous humor that positively labeled for neutrophils as determined by cytometric bead assay.

**Table 1 microorganisms-07-00290-t001:** Oligonucleotide primers for deletion of *ulaR2* by splice overlap extension.

Primer Name	Sequence (5′ to 3′)
spd1961_upF	ACACATCGTAAGGATAGATGCGGA
spd1961_upR	CATCAAGCTTATCGATACCGTTCTCTTCCTTTCTAACTAC
spd1961_downF	GAAGGTTTTTATATTACAGCTCCAATGTTAAAAATTGGTACAGC
spd1961_downR	TATTTTTGCACAAATGCTGGGGA

**Table 2 microorganisms-07-00290-t002:** *S. pneumoniae* D39 genes essential for growth in rabbit vitreous humor.

Gene	Description	Reduction in Fitness	*p*-value ^1^
*spd_0006*	Transcription repair coupling factor	86.06%	0.026
*spd_0095*	Conserved hypothetical protein	7.42%	0.022
*spd_1004*	Glyceraldehyde-3-phosphate dehydrogenase, NADP-dependent	88.53%	0.016
*spd_1225*	Conserved hypothetical protein	38.43&	0.014
*spd_1489*	N-acetylneuraminate lyase, putative	11.92%	0.028
*spd_1846*	Phosphotransferase system IIB component	65.13%	0.020

^1^ Values indicate a comparison of sequencing reads between libraries grown in Todd Hewitt broth containing yeast extract (THY) and vitreous humor.

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
