# Peer review of "A Transcriptional Activator of Ascorbic Acid Transport in Streptococcus pneumoniae Is Required for Optimal Growth in Endophthalmitis in a Strain-Dependent Manner"

_microorganisms, 2019, doi:10.3390/microorganisms7090290_

Round 1

Reviewer 1 Report

Peer review of “A transcriptional activator of ascorbic acid transport in Streptococcus pneumoniae is required for optimal growth in endophthalmitis in a strain-dependent manner,” by Benton, et al, of consideration for publication in the journal Microorganisms.

In this study the authors used Tn-Seq to analyze genes required for Streptococcus infection inside the eye. A number of genes were found to contribute to fitness within the eye. Of these spd_1846 was chosen for further analysis. This gene was replaced in two strains using homologous recombination, and the resulting mutants were evaluated using a rabbit endophthalmitis infection model.  The importance of UlaR2 appeared to be strain dependent, but indicates UlaR2 as a new contributor to ocular infection.

Overall, the manuscript was well written and put together, with some exceptions noted below.  The study is appropriate for this journal, and would be excellent for the upcoming issue on ocular infections and ocular microbiology. The study has high novelty due to the use of  RNA-seq for an ocular infection model.  The major negative is that there are several points at which additional textual information or analysis should be provided to improve the manuscript.

Line 96. Please cite Himar transposon.

Lines 89-95. Please provide more data for the tn-sequencing experiment. How many mutant bacteria were used, etc.

Table 2. Please indicate the relative reduction in fitness of the isolate mutants.

Please graphically represent represent the operon containing Spd_1846.

Line 198. Consider cutting “likewise”, and indicate that Spd_1846 was deleted and replaced with a chloramphenicol resistance cassette. If a transcriptional stop was included in the cassette, please indicate whether this is so.

Line 215. Please note whether the 5 base pair changes are expected to result in amino acid changes in the protein.

Figure 1. Please place hash marks on the x-axis to indicate the change in scale between 12 and 24 hours.

Line 231. Cut “target of”

Line 236. Cite or define the evaluation criteria. Please indicate whether any mutants with increased fitness were isolated.

Author Response

We thank this reviewer for the careful review and suggestions to improve this manuscript. We have provided responses to each item below, and have used the Track Changes tool in Word to show the changes.  Thank you.

Line 96. Please cite Himar transposon.

Response: We have cited the transposon as well as clarified that it is a derivative of Himar.

Lines 89-95. Please provide more data for the tn-sequencing experiment. How many mutant bacteria were used, etc.

Response: We have added more information. We used a previously published protocol that has extensive details in it (reference 22) and reiterated its citation as well.

Table 2. Please indicate the relative reduction in fitness of the isolate mutants.

Response: This has been added to the table.

Please graphically represent represent the operon containing Spd_1846.

Response: We have done as requested and included it as the new Figure S1. We tried to make the figure distinct enough from that which has already been published by Afzal et al. [29].

Line 198. Consider cutting “likewise”, and indicate that Spd_1846 was deleted and replaced with a chloramphenicol resistance cassette. If a transcriptional stop was included in the cassette, please indicate whether this is so.

Response: We have done as requested.

Line 215. Please note whether the 5 base pair changes are expected to result in amino acid changes in the protein.

Response: We have done as requested.

Figure 1. Please place hash marks on the x-axis to indicate the change in scale between 12 and 24 hours.

Response: We have done as requested.

Line 231. Cut “target of”

Response: We have done as requested.

Line 236. Cite or define the evaluation criteria. Please indicate whether any mutants with increased fitness were isolated.

Response: We have done as requested.

Reviewer 2 Report

The authors present a well-written and nicely organized manuscript titled “A Transcriptional Activator of Ascorbic Acid 2 Transport in Streptococcus pneumoniae is Required 3 for Optimal Growth in Endophthalmitis in a Strain-4 dependent Manner”. This study demonstrates ascorbic acid transcriptional activator is essential for growth of S. pneumoniae D39 strain in both in vitro and in vivo model and E335 strain in invitro model. Overall, I enjoyed reading this manuscript, though before publication I have the following recommendations for clarity and completeness.

Was there a transposon library for E335 available? if it is available, what is the reason for authors not utilizing that for transposon mutant library analysis?

Does the authors have any possible reason of why E335 growth effected in invitro but not in in vivo with deletion of “ulaR2”, in my opinion this should be explored a little bit more in discussion.

Minor comment:

Line 309: change “E33 to E335”.

Author Response

We thank this reviewer for the careful review and suggestions to improve this manuscript. We have provided responses to each item below, and have used the Track Changes tool in Word to show the changes.  We hope that our answers have alleviated any concerns. Thank you.

Was there a transposon library for E335 available? if it is available, what is the reason for authors not utilizing that for transposon mutant library analysis?

Response: No, there was not a library available but we are currently working on it. It will be very interesting to determine the fitness differences between strains once we can assess the E335 library.

Does the authors have any possible reason of why E335 growth effected in invitro but not in in vivo with deletion of “ulaR2”, in my opinion this should be explored a little bit more in discussion.

Response: We have added a paragraph (second paragraph) in the Discussion.  Thank you very much for pointing this out.

Minor comment:

Line 309: change “E33 to E335”.

Response: We have fixed this error.

Reviewer 3 Report

In this paper, authors have undertaken a series of experiments and shown that an isogenic gene deletion mutant of the ascorbic acid transcriptional activator, ulaR2, in 2 strains of S. pneumoniae. Although the imvolvement of ascorbic acid transportation in bacterial cells has been reported in other papers, authors showed that ulaR2 is necessary for growth in D39 strain but not in E335. Results show that ulaR2 may be important for fitness during S. pneumoniae endophthalmitis depending on the background of the strain. The methodology is appropriate, results clear and conclusions appropriate. However, following discussion should be described in the manuscript.

In in vitro study, both D39 and E335 need ulaR2 for their growth. On the other hands, only D39 needs ulaR2 in rabbit vitreous. What is the critical difference between THY medium and rabbit vitreous fluid for the growth.

Authors used rabbit vitreous humor in the study. How do they think the difference derived from species? Especially, is there any speculation about human endophthalmitis?

Author Response

We thank this reviewer for the careful review and suggestions to improve this manuscript. We have provided responses to each item below, and have used the Track Changes tool in Word to show the changes.  We believe the discussion has been improved based on this reviewer’s comments. Thank you.

In in vitro study, both D39 and E335 need ulaR2 for their growth. On the other hands, only D39 needs ulaR2 in rabbit vitreous. What is the critical difference between THY medium and rabbit vitreous fluid for the growth.

Response: We have added a new paragraph to the Discussion (second paragraph) to discuss this topic and hope that it is satisfactory for this reviewer. Thank you for pointing out this topic that needed discussion.

Authors used rabbit vitreous humor in the study. How do they think the difference derived from species? Especially, is there any speculation about human endophthalmitis?

Response: This is a topic that we find both interesting and challenging. There are multitudes of components of the vitreous, and we could not find literature that compared the two species in depth outside of pharmacokinetics measures of drug clearance. We think that, if bacterial targets are narrowed down in animals, then in the future we could justify using explanted human vitreous humor to test these targets.  We have added a couple of sentences discussing the issue of rabbit versus human at the end of the Discussion.